# Triangular lattice quantum dimer model with variable dimer density

Zheng Yan [1], Rhine Samajdar [2], Yan-Cheng Wang[3], Subir Sachdev [2,4] & Zi Yang Meng [1]

Quantum dimer models are known to host topological quantum spin liquid phases, and it has recently become possible to simulate such models with Rydberg atoms trapped in arrays of optical tweezers. Here, we present large-scale quantum Monte Carlo simulation results on an extension of the triangular lattice quantum dimer model with terms in the Hamiltonian annihilating and creating single dimers. We find distinct odd and even $\mathbb{Z}_2$ spin liquids, along with several phases with no topological order: a staggered crystal, a nematic phase, and a trivial symmetric phase with no obvious broken symmetry. We also present dynamic spectra of the phases, and note implications for experiments on Rydberg atoms.

Recent quantum simulation advances have provided remarkable microscopic access to the quantum correlations of a $\mathbb{Z}_2$ quantum spin liquid (QSL)[1,2]. The $\mathbb{Z}_2$ QSL[3,4] is the simplest quantum state in two spatial dimensions with fractionalized excitations and time-reversal symmetry, and has the same anyon content as the toric code[5]. Once we include considerations of lattice and other symmetries, $\mathbb{Z}_2$ QSLs come in different varieties; the distinctions between them are important in understanding the phase diagrams of possible experimental realizations. The coarsest classification subdivides $\mathbb{Z}_2$ QSLs into "odd" and "even" classes, depending upon whether elementary translations anticommute or commute when acting on excitations carrying $\mathbb{Z}_2$ magnetic flux[6–9], and results in different translational symmetry fractionalization patterns and spectral signatures in the dynamic response[10–15]. More refined classifications have been obtained since[16–20].

Quantum dimer models (QDMs)[21,22] on nonbipartite lattices have long been known to host $\mathbb{Z}_2$ QSLs. In this work, we investigate an important—but hitherto unexplored—extension of the quantum dimer model on the triangular lattice[23–25]. Unlike the more conventionally studied QDMs, here, the density of dimers is allowed to vary by terms in the Hamiltonian which can annihilate and create single dimers on each link of the triangular lattice. Such a dimer-nonconserving term is motivated by connections to models of ultracold atoms trapped in optical tweezers[26,27], in which each dimer is identified with an atom excited to a Rydberg state by laser pumping[28–30]. The observations of

ref. 2 are for the case where the atoms are positioned on the *links* of the kagome lattice; this connects to the quantum dimer model on the *kagome* lattice[29]. Our study pertains to the triangular-lattice dimer model, which connects to the case where the atoms are placed on the *sites* of the kagome lattice[24,25,28]; such a configuration can be readily realized in the experiments, and initial explorations of quantum phases in such a lattice have already been carried out by the team of ref. 2.

With a dimer-nonconserving term present, here we show, the triangular-lattice quantum dimer model displays novel features relevant to the Rydberg-atom experiments. When the nonconserving terms are large, we can obtain a 'trivial' phase with neither topological order nor broken lattice symmetry. More interestingly, the phase diagram of this extended QDM also harbors both odd and even $\mathbb{Z}_2$ liquids. Note that in early discussions of such QSLs in dimer models, the distinction between the liquids was tied to whether the number of dimers on each site was constrained to be odd or even[24,25]. In the present model, the number of dimers on each site fluctuates between odd and even values, namely 1 and 2; nevertheless, the distinction between even and odd QSLs still survives based on the symmetry transformation properties of excitations with magnetic $\mathbb{Z}_2$ flux ("visons"). In the case with a dimer number constraint on each site, there is an anomaly relation requiring that odd (even) dimers produce vison translations that anticommute (commute)[18,19]. However, in the case without a dimer number constraint (or a soft constraint), of interest to us here, microscopic details will determine whether vison

[1]Department of Physics and HKU-UCAS Joint Institute of Theoretical and Computational Physics, The University of Hong Kong, Pokfulam Road, Hong Kong SAR, China. [2]Department of Physics, Harvard University, Cambridge, MA 02138, USA. [3]Beihang Hangzhou Innovation Institute Yuhang, Hangzhou 310023, China. [4]School of Natural Sciences, Institute for Advanced Study, Princeton, NJ 08540, USA. ✉e-mail: sachdev@g.harvard.edu; zymeng@hku.hk

translations anticommute or commute, and we will investigate this fate numerically with quantum Monte Carlo simulations.

Finally, our study also obtains several phases which break lattice symmetries, but are topologically trivial. This includes two "staggered" phases[23], a "columnar" phase[31], and a "nematic" phase[24,25,32], and we also discuss their density-wave-ordered counterparts in the context of experiments on Rydberg quantum simulators.

## Results

### The model

We investigate the following general dimer Hamiltonian, with one or two dimer(s) per site, on the triangular lattice,

$$
\begin{aligned}
H = \quad & -t \sum_r \left( \left| \rotatebox{45}{\rule{0pt}{0pt}} \right\rangle \left\langle \rule{0pt}{0pt} \right| + \text{h.c.} \right) \\
& +V \sum_r \left( \left| \rule{0pt}{0pt} \right\rangle \left\langle \rule{0pt}{0pt} \right| + \left| \rule{0pt}{0pt} \right\rangle \left\langle \rule{0pt}{0pt} \right| \right) \\
& -h \sum_l \left( \left| \rule{0pt}{0pt} \right\rangle \left\langle \rule{0pt}{0pt} \right| + \text{h.c.} \right) \\
& -\mu \sum_l \left( \left| \rule{0pt}{0pt} \right\rangle \left\langle \rule{0pt}{0pt} \right| \right),
\end{aligned} \tag{1}
$$

where the sum on $r$ runs over all plaquettes (rhombi), including the three possible orientations, and $l$ runs over all links. The different terms in this Hamiltonian are as follows. The kinetic term (controlled by $t$) flips the two dimers on every flippable plaquette, i.e., on each plaquette with two parallel dimers, while the potential term (controlled by the interaction $V$) describes a repulsion ($V > 0$) or an attraction ($V < 0$) between nearest-neighbor dimers. The transverse-field term of strength $h$ creates/annihilates a dimer at link $l$ (similar terms also appear in the quantum realization of the classical models of ref. [32]), in contrast to the $t$ and $V$ terms, neither of which change the dimer number. Lastly, $\mu$ sets the chemical potential for the occupation of a link by a dimer. We further impose a soft constraint requiring that there must be one or two dimer(s) per site. Thus, when $\mu \to \pm\infty$, the model reverts to the conventional hard-constrained quantum dimer model with exactly two or one dimer(s) per site—the phase diagrams of both these QDMs have been extensively studied in the literature[24,25,31,33-37]. Hereafter, we set $t = 1$ as the unit of energy for the rest of this paper.

To solve the model in Eq. (1) in an unbiased manner, we employ the recently developed sweeping cluster quantum Monte Carlo algorithm, which can perform efficient sampling in constrained quantum many-body systems[37-40]. By monitoring the behavior of various physical observables such as dimer correlation functions and structure factors, we map out the detailed phase diagrams, such as, for instance, in Fig. 1. Moreover, in addition to static observables, we also compute the dynamic dimer correlation functions in imaginary time and employ the stochastic analytic continuation method[12,13,37,41-46] to obtain the dynamic dimer spectral functions in real frequencies. Our simulations are performed on the triangular lattice with periodic boundary conditions and system sizes $N = 3L^2$ for linear dimensions $L = 8, 12, 16, 18, 24$, while setting the inverse temperature $\beta = L$ ($\beta = 200$) for equal-time (dynamical) simulations.

### The phase diagram

Although the phase diagrams in the two limits with exactly 1/3 and 1/6 dimer fillings are well understood, the manner in which they connect to each other in the presence of a nonzero transverse field $h$ and chemical potential $\mu$ is an interesting open question. In particular, one may ask what happens between the two kinds of $\mathbb{Z}_2$ QSLs, i.e., whether they are separated by a direct phase transition or an intermediate phase. An important reason this question has remained unaddressed so far is the lack of a suitable algorithm to deal with the soft constraint. As discussed in detail in the section in "Methods", here, we adapt the sweeping cluster Monte Carlo algorithm used for hard-constrained QDMs[38,39] to soft ones and use it to map out the phase diagram of the Hamiltonian in Eq. (1). Figure 1 shows the full phase diagram obtained at $h = 0.4$, which we focus on in the main text, leaving the discussion of similar phase diagrams with different $h$ to Supplementary Notes 2 and 3 of the Supplementary Information (SI).

The phase diagram exhibits four different symmetry-breaking phases, including the nematic, the columnar, and two staggered phases; the schematic plots of these crystalline phases are shown in the right panels of Fig. 1. Furthermore, we observe two distinct $\mathbb{Z}_2$ QSL phases, which are denoted as "Even QSL" and "Odd QSL" in the figure. In addition, a trivial disordered—or paramagnetic (PM)—phase exists in the central region in between the two QSLs; note that such a PM phase

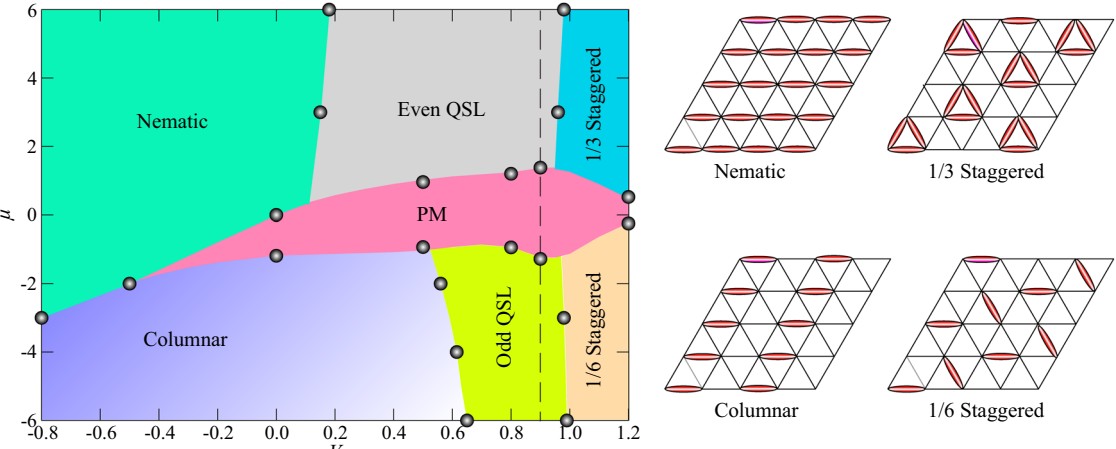

**Fig. 1 | Phases of the variable-density triangular lattice QDM.** Left panel: The full phase diagram, spanned by the $V$ and $\mu$ axes, is obtained from QMC simulations at $h = 0.4$. The phase boundaries between the paramagnetic (PM) phase and the two QSLs along the dashed line are studied in Fig. 3; the phase transitions are first-order. The phase boundaries between the QSLs and the nematic, columnar, and staggered phases are shown in Supplementary Note 3 of the Supplementary Information (SI). The associated transitions are either continuous (such as the QSL−nematic and QSL−columnar) or first-order (such as the QSL--staggered). Right panel: Schematic pictures of the four crystalline phases (nematic, columnar, 1/3 staggered, and 1/6 staggered). In the limit of exactly one dimer per site, a $\sqrt{12} \times \sqrt{12}$ valence bond solid (VBS) phase is known to exist between the odd QSL and the columnar phase. However, it is nearly degenerate with the columnar phase over a large region in our simulations, and we depict this schematically by using a lighter shading for the columnar phase near the odd QSL.

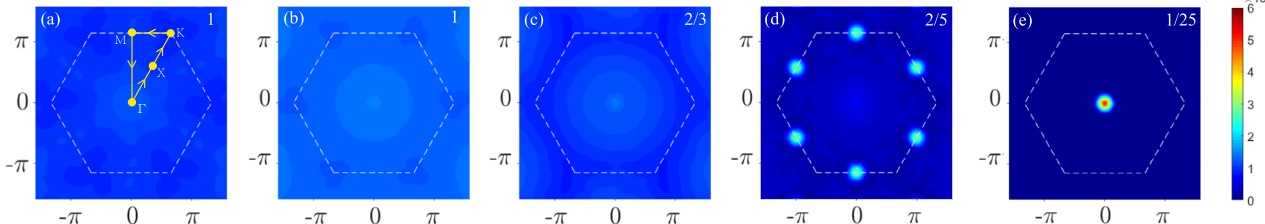

**Fig. 2 | Equal-time dimer-structure factors.** Here, we present $D(\mathbf{k}, \tau = 0)$ in the Brillouin zone for the (**a**) odd $\mathbb{Z}_2$ QSL ($\mu = -3$, $V = 0.9$), **b** PM phase ($\mu = 0$, $V = 0.9$), **c** even $\mathbb{Z}_2$ QSL ($\mu = 3$, $V = 0.9$), **d** columnar phase ($\mu = -3$, $V = -0.5$), and **e** nematic phase ($\mu = 3$, $V = -0.5$) in the phase diagram of Fig. 1. All the data are simulated using $\beta = L = 12$. The upper-right labels in each panel represent the scaling factor for the intensities such that the five panels can be scaled onto the same color bar. In addition, the high-symmetry path for the spectra in Fig. 4 is also drawn in (**a**).

does not arise in the more familiar QDMs where the dimer number per site is exactly constrained. The phase boundaries between these phases are determined by examining various parameter points and paths scanning through the phase diagram, such as the dashed line in Fig. 1.

To characterize this rich variety of phases, we compute the equal-time ($\tau = 0$) dimer-structure factor (see Fig. 2) as

$$D(\mathbf{k},\tau) = \frac{1}{N} \sum_{\substack{i,j \\ \alpha = 1,2,3}}^{L^3} e^{i\mathbf{k}\cdot\mathbf{r}_{ij}} \left( \langle n_{i,\alpha}(\tau) n_{j,\alpha}(0) \rangle - \langle n_{i,\alpha} \rangle \langle n_{j,\alpha} \rangle \right), \quad (2)$$

where $n_i$ is the dimer number operator on bond $i$ and $\alpha$ stands for the three bond orientations, at five representative parameter points corresponding to the five different phases in the phase diagram. Figure 2a–c shows $D(\mathbf{k}, 0)$ inside the odd QSL, PM, and even QSL phases, respectively. In the hexagonal Brillouin zone, we observe that there are no peaks associated with long-range order but only broad profiles signifying different short-range dimer correlation patterns in real space. In contrast, Fig. 2d, e presents the dimer-structure factors inside the columnar and nematic phases, respectively. One now clearly sees the Bragg peaks at the $M$ points for the columnar phase (there can be three different orientations of the columnar dimers, corresponding to all the three pairs of $M$ points), and at the $\Gamma$ point in the nematic phase.

### The two $\mathbb{Z}_2$ QSLs

Having established the lack of long-range dimer–dimer correlations in the odd/even $\mathbb{Z}_2$ QSLs and the PM phase, next, we move on to the phase transitions between them. Since all three of these phases are disordered, care needs to be taken in determining their phase boundaries. Our results in this regard are summarized in Fig. 3, which shows the data along a path with a fixed $V = 0.9$ and varying $\mu$ in the phase diagram (dashed line in Fig. 1).

First, in Fig. 3a, we illustrate the energy density curves, which appear to be smooth without any obvious turning points along the path as $\mu$ is scanned. However, when the transverse field becomes large, we expect that all the links should be polarized along the $x$ axis (if there were no constraints). Since the model in Eq. (1) can be regarded as a spin model with spins on links (occupied/empty links being equivalent to spin up/down), the polarization

$$M_x = \frac{1}{N} \sum_l \left( \left| \begin{array}{c} \rule{0.8em}{0.15em} \end{array} \right\rangle \left\langle \begin{array}{c} \bullet\!-\!\bullet \end{array} \right| + \text{h.c.} \right) \sim \frac{1}{N} \sum_l S_l^x$$

can be used to describe the level of polarized links (spins), and thus, to probe the PM phase. Indeed, as seen in Fig. 3b, $M_x$ helps us to identify a first-order phase transition between the PM phase and the two $\mathbb{Z}_2$ QSLs. In the PM phase, $M_x$ becomes large but is still far from the classical saturation value of 1; this is because the soft constraint forbids

all links from being fully polarized simultaneously. We can also discover similar first-order phase transitions, at the same parameter points, independently from the dimer filling $\rho$ shown in Fig. 3c. In the even (odd) $\mathbb{Z}_2$ QSL phase, the filling is nearly 1/3 (1/6) while the filling changes continuously in the PM phase.

In addition, a closed string operator[2], schematically defined as in Fig. 3e, f as $\langle string \rangle = \langle (-1)^{\#\text{cut dimers}} \rangle$ on a rhomboid with odd linear size, can be used to distinguish the two QSLs and the PM phase. As shown in Fig. 3e, f, $\langle string \rangle$ should be ±1 in a *pure* even/odd $\mathbb{Z}_2$ QSL without spinons and 0 in a PM phase. We measure all the $3 \times 3$ rhomboids in the lattice to obtain the expectation value $\langle string \rangle$ along the path scanning $\mu$ at $V = 0.9$. The resultant data in Fig. 3d indeed reveal that inside the odd (even) $\mathbb{Z}_2$ QSL phase, $\langle string \rangle \sim -1$ ($\langle string \rangle \sim 1$), while inside the PM phase, $\langle string \rangle \approx 0$; the transitions are also seen to be first-order, in consistency with Fig. 3b, c.

### The dynamical dimer spectra

One of the hallmarks of a QSL is its ability to support fractionalized excitations that cannot be created individually by any local operator. In this section, we focus on one class of such fractional excitations with magnetic $\mathbb{Z}_2$ flux, i.e., the visons. Naturally, vison configurations with different fluxes will result in different dimer spectral signatures, thus realizing, in particular, the interesting phenomenon of translational symmetry fractionalization[10–13,37], which can be further used to distinguish the PM and the even/odd $\mathbb{Z}_2$ QSLs and make a possible connection to experiments. To this end, we compute the dimer spectra, obtained from stochastic analytic continuation of the Monte Carlo-averaged dynamic dimer correlation function $D(\mathbf{k}, \tau)$ with $\tau \in [0, \beta]$ (which can be viewed as the dynamical vison-pair correlation functions deep inside the $\mathbb{Z}_2$ QSLs[37]; more details can be found in the Supplementary Note 1). Figure 4a shows that in the odd $\mathbb{Z}_2$ QSL phase, the gapped dimer (vison-pair) spectrum forms a continuum, and the dispersion minima are located at both the $M$ and $\Gamma$ points[35,37]. On the other hand, Fig. 4c illustrates that the dimer (vison-pair) spectrum deep inside the even $\mathbb{Z}_2$ QSL is also a continuum but with minima only at $\Gamma$. These features are consistent with the expectation that the visons of the odd $\mathbb{Z}_2$ QSL carry a fractional crystal momentum, whereas visons of the even QSL do not[12,37]. Note that for the single vison dispersion of an odd QSL, the locations of the minima are dependent on the chosen gauge[30,47,48] whereas the vison-pair spectrum is a gauge-invariant observable. For the even QSL, refs. 24,25 found that the minima of the mean-field vison dispersion occur at the three inequivalent $M$ points in the Brillouin zone. Accordingly, one would then expect the vison-pair spectrum to exhibit a minimum at $\Gamma$ (which is equivalent to $2M$ modulo a reciprocal lattice vector), in agreement with our numerical results. The arguments above apply generally to the dynamics of an odd/even QSL and should hold even at finite $\mu$; similar behaviors have also been observed for the odd/even QSLs of the Balents–Fisher–Girvin (BFG) model[12]. In comparison, Fig. 4b presents the dimer spectrum inside the PM phase; here, there exists no clear continuum in the frequency

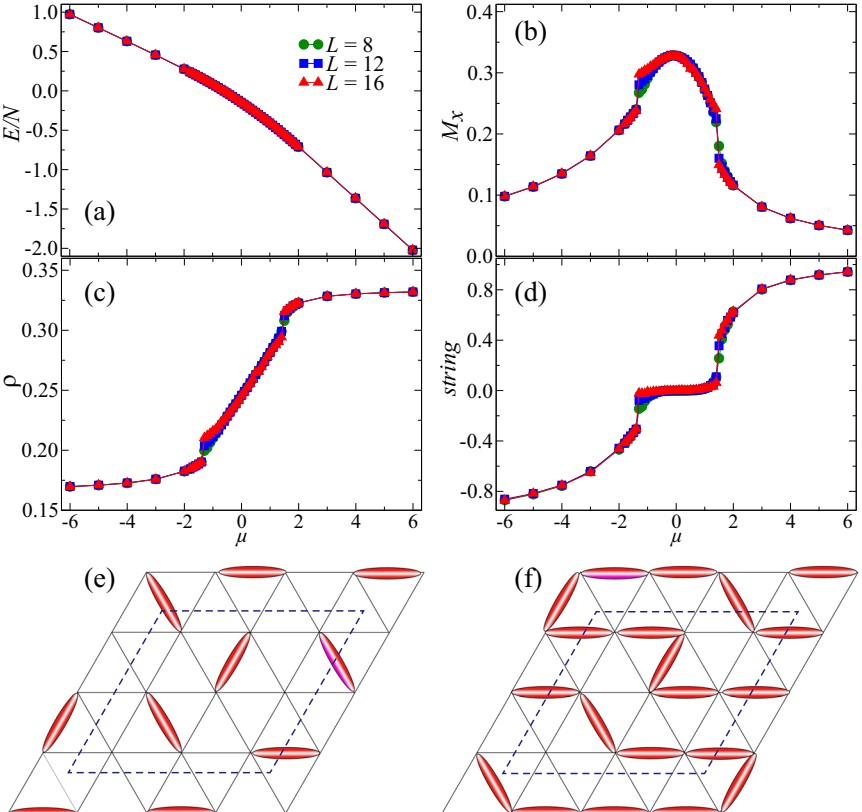

**Fig. 3 | Phase transitions between QSLs and the PM phase.** Data along the QSL-PM-QSL path, indicated by the dashed line at $V = 0.9$ in Fig. 1. **a** The energy density is smooth with increasing $\mu$. **b** The polarization $M_x$ reveals the first-order phase transition between the PM phase and the two $\mathbb{Z}_2$ QSLs. **c** The dimer filling remains at approximately $\rho = 1/3$ in the even QSL and $\rho = 1/6$ in the odd QSL. It changes continuously in the PM phase, and the filling also exhibits a first-order phase transition between the PM phase and QSLs. **d** The string operator is zero in the trivial PM phase but positive (negative) in the even (odd) $\mathbb{Z}_2$ QSL. All the data are calculated for $V = 0.9$, $\beta = L$, $h = 0.4$. **e** In a pure odd $\mathbb{Z}_2$ QSL with dimer filling $\rho = 1/6$, a string operator defined on a rhomboid with odd linear size (3 in this case) should attain the value −1. **f** In a pure even $\mathbb{Z}_2$ QSL with dimer filling $\rho = 1/3$, the string operator should always yield 1. The string operators presented in (**d**) are measured for a $3 \times 3$ rhombus averaged over the entire lattice for different $L$.

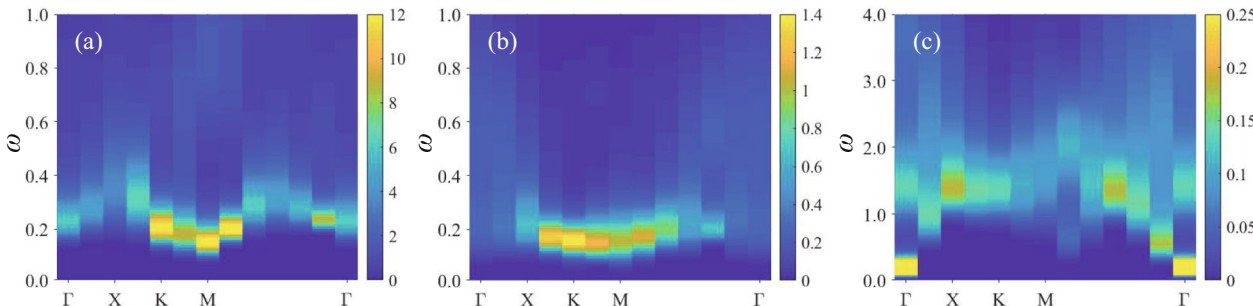

**Fig. 4 | Dynamical dimer spectra.** The dimer spectra in the (**a**) odd $\mathbb{Z}_2$ QSL in the limit of one dimer per site, corresponding to $\mu \to -\infty$ and $V = 1$ in Fig. 1, **b** PM phase with $\mu = 0$, $V = 0.9$ and $h = 0.4$, and **c** even $\mathbb{Z}_2$ QSL in the limit of two dimers per site, corresponding to $\mu \to \infty$ and $V = 0.5$ in Fig. 1. The dimer spectra exhibit continua in both (**a**) and (**c**), conveying the fractionalization of spins into visons. However, the dispersion minima in the two cases differ, being located at both $M$ and $\Gamma$ for (**a**) and only at $\Gamma$ for (**c**), representing the translational symmetry fractionalization in the former and the lack thereof in the latter. In (**b**), however, the dimer spectrum is flat and displays less of a continuum in the frequency domain, consistent with a polarized PM phase. All the data are simulated at $\beta = 200$ on a $L = 12$ lattice, with the low-temperature $T = 1/200$ being necessary to overcome the small vison gap and the transverse field $h$.

domain, indicating the lack of fractionalization of dimers into pairs of visons. Moreover, the overall dispersion is flat, which is consistent with the dispersionless $S^z$ spectrum in an $S^x$-polarized state, such as in the transverse-field Ising model.

## Discussion
In this work, we investigate a QDM with variable dimer density on the triangular lattice and uncover a plethora of interesting phases, including crystalline solids and two distinct classes of highly entangled QSL states hosting fractionalized excitations. Through detailed quantum Monte Carlo analyses, we explore the subtle interplay between these different phases and find the unique properties of their static and dynamic fingerprints. With the remarkable advances in quantum simulation, experimental realization of the dimer model in Eq. (1) should provide new probes of novel QSL phases and their phase transitions.

**Fig. 5 | Update scheme of soft constraint.** For the soft constraint of 1 or 2 dimer(s) per site, we have to consider all the neighbors when creating/annihilating a dimer on the central link. **a** When both the A and B sites have one dimer, one is allowed to create/annihilate a dimer on the center link. **b** It is forbidden to create a dimer on the center link when either the A or the B site already has two dimers. **c** It is forbidden to annihilate a dimer on the center link when either the A or the B site has only one dimer.

In particular, our results could find application to recent experiments with programmable quantum simulators based on highly tunable Rydberg-atom arrays, which have emerged as powerful platforms to study strongly correlated phases of matter and their dynamics. While our extended QDM differs from models of Rydberg atoms on the sites of the kagome lattice[28] in the precise form of the $V$ interactions, the two systems bear resemblance in some of their phases. Specifically, the Rydberg model also displays the 1/6 staggered and nematic phases of Fig. 1, separated by a 'liquid' regime with no broken symmetry. These ordered phases can be mapped to the solid phases of a triangular-lattice QDM with either one or two dimers per site, which precisely constitutes our soft constraint. Appealing to the universality of phase transitions[28], possible fates of the liquid state in the Rydberg model are then one or more of the phases obtained by interpolating between the 1/6 staggered and nematic phases in Fig. 1 for the present quantum dimer model: namely, the odd QSL, the PM, and the even QSL. These considerations highlight the potential utility of variable-density dimer models in the experimental realm and provide a pathway to studying their rich physics.

## Methods
### Sweeping cluster algorithm
This is a quantum Monte Carlo method developed by the authors to solve the path integral of constrained quantum many-body models[37–40,49]. The key idea of the sweeping cluster algorithm is to sweep and update layer by layer along the imaginary-time direction, so that the local constraints (gauge fields) are recorded by update lines. In this way, all the samplings are performed in the restricted Hilbert space, i.e., the low-energy space. The original sweeping cluster QMC method[38,39] is designed for hard-constraint models, i.e., models in which the number of dimer(s) per site is fixed[37,50]. To solve our models in this work, we further improve upon the prior methods to be able to simulate a soft-constrained dimer model.

The Hamiltonian that we consider is given by Eq. (1) supplemented with the "soft" constraint that there can only be either one or two dimer(s) per site. The definition of winding numbers[39,51–54] for these two cases are explained in Supplementary Note 3.

Similar to the practice in Stochastic Series Expansion types of quantum Monte Carlo methods[55], we separate the Hamiltonian into diagonal and off-diagonal parts. It is obvious that the $t$ and $V$ terms will not change the number of dimer(s) per site, but both the chemical potential $\mu$ and the transverse field term $h$ would. Therefore, the Monte Carlo update will need to obey the soft constraint when we deal with the $\mu$ and $h$ terms. We write the $h$ off-diagonal term and the $\mu$ diagonal term as,

$$H_{d,l} = \mu \left( \left| \rule{1.2em}{0.6ex} \right\rangle \left\langle \rule{1.2em}{0.6ex} \right| \right) + C, \qquad (3)$$

$$H_{o,l} = h \left( \left| \rule{1.2em}{0.6ex} \right\rangle \left\langle \rule[-0.1ex]{0.4ex}{0.4ex} \rule{1em}{0.1ex} \rule[-0.1ex]{0.4ex}{0.4ex} \right| + \text{h.c.} \right), \qquad (4)$$

where $C$ is a constant to ensure that the corresponding matrix elements are positive. The label "$d/o$" indicates whether the operator is

diagonal or off-diagonal, and $l$ labels the links of the lattice. Although these two terms are single-link operators, they may break the soft constraint when considering neighbors, so we have to regard the single-link operator as a multi-link operator instead with all closest neighbors as shown in Fig. 5.

We can design the Monte Carlo algorithm to update vertices according to the soft constraint on the cells as shown in Supplementary Fig. 5. Since the original sweeping cluster method always obeys the constraints without changing the number of dimers per site, adding such considerations for the terms in Eq. (1) into the original sweeping cluster Monte Carlo method makes all samplings satisfy the soft constraint.

### Stochastic analytic continuation
The main idea behind the stochastic analytic continuation (SAC) method[41,42,46,56] is to obtain the optimal solution of the inverse Laplace transform via sampling dependent on the importance of goodness. A set of imaginary-time correlation functions $G(\tau)$ can be obtained through the sweeping cluster QMC method first. The real-frequency spectral function and the imaginary-time correlation function are related by a Laplace transformation as $G(\tau) = \int_0^\infty d\omega (e^{-\tau\omega} + e^{-(\beta-\tau)\omega}) S(\omega)/\pi$. We can inversely solve this equation by fitting a better spectral function. Assume the spectral function has a general form, $S(\omega) = \sum_i a_i \delta(\omega - \omega_i)$. We can obtain the optimal spectral function, i.e., the optimal choice of the set $\{a_i, \omega_i\}$ in the ansatz, numerically through sampling according to the importance of goodness of fit, with a simulated-annealing approach and with respect to the QMC errorbars of the imaginary-time correlation data $G(\tau)$. The reliability of such a QMC-SAC scheme has been extensively tested in various quantum many-body systems, such as the 1D Heisenberg chain[57] compared to the Bethe ansatz, the 2D Heisenberg model[44,58] in comparison to exact diagonalization, field theoretical analysis and neutron scattering spectra in real square-lattice quantum magnets, deconfined quantum critical points[58,59] and deconfined U(1) spin liquid phases with emergent photon excitations[60], $\mathbb{Z}_2$ quantum spin liquid models with fractionalized spectra[12,13,61] via anyon condensation theory, and the quantum Ising model with direct comparison to neutron scattering and NMR experiments[62,63]. We refer the readers to the technical descriptions available in the literature for detailed documentation of our QMC+SAC scheme.

## Data availability
The data that support the findings of this study are available from the authors upon reasonable request.

## Code availability
All numerical codes in this paper are available upon reasonable request to the authors.

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

## Acknowledgements

R.S. and S.S. are supported by the U.S. Department of Energy under Grant DE-SC0019030 and thank their coauthors in earlier collaborations[2,28]. Z.Y. and Z.Y.M. acknowledge support from the Research Grants Counci of Hong Kong SAR of China (Grant Nos. 17303019, 17301420, 17301721, and AoE/P-701/20), the K.C. Wong Education Foundation (Grant No. GJTD-2020-01) and the Seed Funding "Quantum-Inspired explainable-AI" at the HKU-TCL Joint Research Centre for Artificial Intelligence. Y.C.W. acknowledges the support from the NSFC under Grant Nos. 11804383 and 11975024. Y.C.W. and Z.Y. thank the support of Beihang Hangzhou Innovation Institute Yuhang. We thank Beijng PARATERA Tech CO., Ltd., the supercomputing system in the High-performance Computing Centre of Beihang Hangzhou Innovation Institute Yuhang, the HPC2021 system under the Information Technology Services at the University of Hong Kong, and the Tianhe-II platform at the National Supercomputer Center in Guangzhou for their technical support and a generous allocation of CPU time.

## Author contributions

R.S., S.S., and Z.Y.M. initiated the work. Z.Y. developed the QMC algorithm for soft constraint. Z.Y. and Y.C.W. performed the computational simulations. All authors contributed to the analysis of the results. S.S. and Z.Y.M. supervised the project.

## Competing interests

The authors declare no competing interests.
