## [Peer Review File · Nature Communications]

REVIEWER COMMENTS

Reviewer #1 (Remarks to the Author):

In their manuscript, Yan et al. study a very interesting quantum dimer model, where coherent superpositions of different dimer-numbers per site are allowed and controlled by a dimer chemical potential term. As emphasized by the authors, such models are highly relevant to understand ongoing experiments realizing quantum spin-liquids in Rydberg tweezer arrays: In these systems, the dimer number may fluctuate, but such effects are usually not included in conventional quantum dimer Hamiltonians. Hence the work is very timely and addresses a very important question. The main result of the paper is to analyze the phase diagram of their proposed quantum dimer Hamiltonian. They identify different ordered phases (with symmetry breaking), two quantum spin liquids, and — most significantly — find a paramagnetic phase separating the two even / odd Z₂ spin-liquids. The paper is well written, but before I can recommend it for publication, my following questions would have to be clarified:

1) My main concern is about the claim in the manuscript that fractionalization can be observed in the numerically obtained spectra in Fig. 4. In particular, I am not convinced that Fig. 4 a) shows more of a continuum than Fig. 4 b), as claimed in the text. I very much appreciate the numerical efforts by the authors, and I have no doubt that the analytic continuation performed by the authors to extract the spectrum is challenging. But (without further evidence or explanation), I believe the authors should significantly tone down their statement that Fig. 4 shows clear signs of fractionalization.

2) The authors should say more about the nature of the quantum spin liquid they find. They studied the string order parameter effectively probing the emergent Gauss law. Did they also consider other, more informative, measures, such as the Fredenhagen-Marcu string order parameter? I believe the authors argue that the observed fractionalization (see point 1)) signals the existence of free visons — so is their conclusion that the QSLs are in a deconfined regime / topologically ordered? I suppose for the case with infinite (+ or -) chemical potential, the situation is much clearer, but can the authors comment on which signatures they believe remain when considering finite chemical potentials?

3) Can the authors comment on possible experimental implementations (or the experimental relevance) of their constraint requiring that there must be one or two dimers per site (no less and no more)? It would be useful for readers to understand how realistic the model Hamiltonian in Eq. (1) is.

In addition, the following minor points could be addressed:

- In the caption of Fig. 1, the authors mention that a VBS phase is known to exist between the odd QST and the columnar phase, which is nearly degenerate with the columnar phase and hence is conjectured not to show up in the numerically obtained phase diagram. Can this be due to the non-zero (though low) temperature $\beta = L$ considered by the authors?

- It would be very useful if the authors could provide some brief intuition, already in the main text, why the dimer correlation function $D(k, \tau)$ can be viewed as the dynamical vision-pair correlation function, and under which general assumptions this is the case.

- can the authors comment on whether (which) checks they performed that the considered temperatures $\beta = L$ are sufficiently low (is it clear that convergence towards the ground state is achieved)?

Reviewer #2 (Remarks to the Author):

The paper reports a quantum-Monte-Carlo study of a dimer model with a variable dimer density. This model exhibits a rich phase diagram with odd and even Z_2 topological liquids, dimer crystals, and a "paramagnetic phase" (a liquid phase without a topological order). The most interesting feature of the model, in my opinion, is a transition between the even and odd Z_2 liquids via the paramagnetic phase. Such a transition may be prototypical for a wider class of quantum systems, including cold-atom experiments.

I believe that the results are sufficiently interesting for readers in the field of strongly correlated systems in its broad sense (including quantum magnetism, cold atoms, quantum phase transitions, topological order etc.). The methods presented in the paper appear sound, and the numerical results look trustworthy. I would therefore generally recommend this paper for publication in Nature Communications. However, there are a few technical details related to the comparison with earlier results that I would recommend to improve.

1. The possibility of the crystal phase $\sqrt{12} \times \sqrt{12}$ is mentioned in the caption to Fig.1 (and in the supplementary information), but not in the main text. I find it confusing. I believe that this $\sqrt{12} \times \sqrt{12}$ crystal should be mentioned in the main text, together with the references to earlier works where this phase was found. Additionally, I would like to see a more explicit explanation from the

authors on why they don't see this phase in their numerics: if (1) they believe that this phase is not there or (2) their method is in principle not suitable to pick this phase or (3) their method could be suitable to pick this phase, but the energy difference is too small and falls below the sensitivity of their method. Such an explanation would help the reader to evaluate the limitations of the numerics reported in the paper.

2. In the section on the dynamical dimer spectra, where the authors report the minima at the points M and Gamma in the odd-Z2 liquid, it would be worth comparing with the similar spectrum from ref.[35] of the manuscript (A.Ralko et al, 2006), where it was studied for the $h=0$ case.

3. Further in this section, the authors mention earlier works on a single-phonon dispersion in the even-Z2 liquid. For a more balanced presentation, it would also be worth mentioning earlier works on a single-phonon dispersion in the odd-Z2 liquid in the $h=0$ case, specifically [D.Ivanov PRB 70, 094430 (2004)] -- at the RK point $V=t$ and [A.Ralko et al, PRB 76, 140404(R) (2007)] -- away from the RK point.

REPLY TO REVIEWER 1

Reviewer 1: *In their manuscript, Yan et al. study a very interesting quantum dimer model, where coherent superpositions of different dimer-numbers per site are allowed and controlled by a dimer chemical potential term. As emphasized by the authors, such models are highly relevant to understand ongoing experiments realizing quantum spin-liquids in Rydberg tweezer arrays: In these systems, the dimer number may fluctuate, but such effects are usually not included in conventional quantum dimer Hamiltonians. Hence the work is very timely and addresses a very important question. The main result of the paper is to analyze the phase diagram of their proposed quantum dimer Hamiltonian. They identify different ordered phases (with symmetry breaking), two quantum spin liquids, and — most significantly — find a paramagnetic phase separating the two even / odd \mathbb{Z}_2 spin-liquids. The paper is well written, but before I can recommend it for publication, my following questions would have to be clarified:*

Reply: We thank the reviewer for their positive assessment of our work and for their insightful comments below, which actually inspired us to improve the manuscript. We reply to the comments one by one now.

Reviewer 1: *1) My main concern is about the claim in the manuscript that fractionalization can be observed in the numerically obtained spectra in Fig. 4. In particular, I am not convinced that Fig. 4 a) shows more of a continuum than Fig. 4 b), as claimed in the text. I very much appreciate the numerical efforts by the authors, and I have no doubt that the analytic continuation performed by the authors to extract the spectrum is challenging. But (without further evidence or explanation), I believe the authors should significantly tone down their statement that Fig. 4 shows clear signs of fractionalization.*

Reply: We acknowledge that the presentation of Fig. 4 could have initially been misleading. From the color bar, one can see that the spectral weight of Fig. 4(a) is much larger than that of (c), which makes the continuum of (a) seem weaker than in (c). This is because the significant weight at the M point in (a) obfuscates the continuum. If we set the threshold of the color bar lower in Fig. 4(a) of the main text, we obtain Fig. R1, which clearly shows much more of a continuum than in (b). Moreover, the strong weight at the M point also reveals that the visons carry π Berry phase in the odd \mathbb{Z}_2 spin liquid and will be bound at both Γ (0,0) and M ($\pi,0$) momentum points.

Reviewer 1: *2) The authors should say more about the nature of the quantum spin liquid they find. They studied the string order parameter effectively probing the emergent Gauss law. Did they also consider other, more informative, measures, such as the Fredenhagen-Marcu string order parameter?*

Reply: We thank the referee for for the suggestion. The Fredenhagen-Marcu string order parameter is very difficult to calculate with quantum Monte Carlo methods. As the definition by K Gregor *et al.* *New J. Phys.* 13, 025009 (2011) shows, the Fredenhagen-Marcu string order parameter is given by $R(L) = \langle F|ss'\rangle / \sqrt{\langle ss'|ss'\rangle}$ with $|ss'\rangle = \tau_s^z \tau_{s'}^z \prod_{l \in C_{ss'}} \sigma_1^z(-T/2)|G\rangle$, where σ_1 are the gauge fields living on the links, and τ_s are the matter fields residing on the sites. However, measuring $\sigma_1^z(-T/2) = e^{-HT/2} \sigma_1^z e^{-HT/2}$ is presently impossible (for us at least) with Monte Carlo simulations.

On the other hand, we have tried to distinguish the between the QSL and PM phases via several different

FIG. R1: Dynamical dimer spectra of Fig. 4(a) plotted using a different color scale.

measurements. In addition to the string operator, we also observed the characteristics of the first-order phase transition clearly from the dimer density, ρ , and the transverse-field-polarized spins, M_x , as Fig. 3 illustrates, thus confirming the phase transition between the QSL and PM phases.

Reviewer 1: I believe the authors argue that the observed fractionalization (see point 1) signals the existence of free visons — so is their conclusion that the QSLs are in a deconfined regime / topologically ordered? I suppose for the case with infinite (+ or -) chemical potential, the situation is much clearer, but can the authors comment on which signatures they believe remain when considering finite chemical potentials?

Reply: We agree with the reviewer’s understanding on what we want to show via the spectra. Owing to the high numerical requirements for accuracy of the analytical continuation technique, it is challenging to obtain good spectra at finite chemical potential. Thus, we have presented arguments about the spectra in the limiting cases. However, we believe the physics should be similar at finite chemical potential, deep within the QSL phases at least, and we have added some discussion along these lines in the manuscript.

Reviewer 1: 3) Can the authors comment on possible experimental implementations (or the experimental relevance) of their constraint requiring that there must be one or two dimers per site (no less and no more)? It would be useful for readers to understand how realistic the model Hamiltonian in Eq. (1) is.

Reply: We thank the referee for this helpful suggestion. While we have added some comments to the manuscript in this regard, let us also explain the correspondence with the experiments here first. In our previous work [R. Samajdar *et al.*, PNAS. 118, e2015785118 (2021)], we found the most interesting region (between the Nematic phase and the 1/6 Staggered phase) in the phase diagram of Rydberg atoms arrayed on a *kagome* lattice has a continuously varying excitation density from 1/3 to 1/6. This translates to a dimer density that varies between 1 and 2 dimers per site of the *triangular* lattice. Due to the Rydberg blockade, there is a local constraint among dimers, which strongly prohibits more than two dimers per site.

Reviewer 1: In addition, the following minor points could be addressed:

— *In the caption of Fig. 1, the authors mention that a VBS phase is known to exist between the odd QST and the columnar phase, which is nearly degenerate with the columnar phase and hence is conjectured not to show up in the numerically obtained phase diagram. Can this be due to the non-zero (though low) temperature $\beta = L$ considered by the authors?*

Reply: We do not believe that this could be due to the temperature not being low enough. While we have worked hard on distinguishing the $\sqrt{12} \times \sqrt{12}$ VBS and the QSL, a similar problem has been encountered by other researchers as well [see, for instance, A. Ralko *et al.* *Phys. Rev. B* 71, 224109 (2005).] In that paper, the authors showed that the phase transition point between the VBS and QSL depends on the cluster used and is about 0.7 (“type A cluster”) or 0.85 (“type B cluster”). They also found it difficult to obtain the transition via traditional methods—such as looking at the order parameter or the Binder ratio—so they use the energy difference of different topological sectors to label the phase transition point. Since the topological sector is well-defined in the one-dimer-per-site limit, and the $\sqrt{12} \times \sqrt{12}$ VBS and QSL belong to different sectors in certain sizes, they finally obtained the phase transition point via this unconventional scheme. However, the phase transition between the $\sqrt{12} \times \sqrt{12}$ VBS and columnar phases is more difficult to calculate even through this way; what they instead claim is that “a clear columnar pattern appears near $V/t \sim -0.8$ ”. Indeed, we also tried to repeat their result in our previous work [Z. Yan *et al.*, *npj Quantum Materials* 6, 39 (2021)] in the one-dimer-per-site limit, but we also faced similar difficulties.

In the variable-dimer-density case, the problem becomes even harder. As the topological sector is no longer well-defined, we cannot compare the energy difference of the sectors. What we have tried is comparing the energy difference of the QMC results with different initial states. Over a large region, we found that the result will converge to the columnar/ $\sqrt{12} \times \sqrt{12}$ VBS phase if we set the initial state to be the columnar/ $\sqrt{12} \times \sqrt{12}$ VBS, but their energy difference is smaller than the error bar; thus, we cannot conclusively distinguish whether there is a columnar solid or $\sqrt{12} \times \sqrt{12}$ VBS.

Reviewer 1: — It would be very useful if the authors could provide some brief intuition, already in the main text, why the dimer correlation function $D(k, \tau)$ can be viewed as the dynamical vision-pair correlation function, and under which general assumptions this is the case.

Reply: Thanks very much for the suggestion. This point was discussed in our previous work [Z. Yan *et al.* *npj Quantum Materials* 6, 39 (2021)], around Eq. (2). For the current manuscript, we have also added more discussions about this in the Supplementary Material and the main text now.

Reviewer 1: — Can the authors comment on whether (which) checks they performed that the considered temperatures $\beta = L$ are sufficiently low (is it clear that convergence towards the ground state is achieved)?

Reply: Thanks for the comment. In our prior experience, we have found that the relation between β and L is decided by the dynamical exponent z . For a bipartite lattice’s RK point, z is 2, but for nonbipartite lattices, the RK point is gapped. For other continuous phase transition points that may arise in the phase diagram, z is 1 as we know (e.g., the O(4) phase transition between the $\sqrt{12} \times \sqrt{12}$ VBS and the odd QSL, and the O(3) phase transition between the Nematic VBS and the even QSL). Therefore, for finite-size extrapolation, $\beta = L$ is trusted in our case.

REPLY TO REVIEWER 2

Reviewer 2: *The paper reports a quantum-Monte-Carlo study of a dimer model with a variable dimer density. This model exhibits a rich phase diagram with odd and even Z_2 topological liquids, dimer crystals, and a "paramagnetic phase" (a liquid phase without a topological order). The most interesting feature of the model, in my opinion, is a transition between the even and odd Z_2 liquids via the paramagnetic phase. Such a transition may be prototypical for a wider class of quantum systems, including cold-atom experiments.*

I believe that the results are sufficiently interesting for readers in the field of strongly correlated systems in its broad sense (including quantum magnetism, cold atoms, quantum phase transitions, topological order etc.). The methods presented in the paper appear sound, and the numerical results look trustworthy. I would therefore generally recommend this paper for publication in Nature Communications. However, there are a few technical details related to the comparison with earlier results that I would recommend to improve.

Reply: We thank the referee for their careful summary of our work and their positive assessment. In the following, we reply point-to-point to the questions raised.

Reviewer 2: *1. The possibility of the crystal phase $\sqrt{12} \times \sqrt{12}$ is mentioned in the caption to Fig.1 (and in the supplementary information), but not in the main text. I find it confusing. I believe that this $\sqrt{12} \times \sqrt{12}$ crystal should be mentioned in the main text, together with the references to earlier works where this phase was found. Additionally, I would like to see a more explicit explanation from the authors on why they don't see this phase in their numerics: if (1) they believe that this phase is not there or (2) their method is in principle not suitable to pick this phase or (3) their method could be suitable to pick this phase, but the energy difference is too small and falls below the sensitivity of their method. Such an explanation would help the reader to evaluate the limitations of the numerics reported in the paper.*

Reply: We thank the referee for raising this good question and regret the confusion. We have now explained this point in the updated manuscript. The primary difficulty is exactly the point (3) mentioned by the referee, i.e., it is very hard for our simulation to distinguish the energy difference between the $\sqrt{12} \times \sqrt{12}$ VBS and columnar phases.

While we have worked hard on distinguishing the $\sqrt{12} \times \sqrt{12}$ VBS and the QSL, a similar problem has been encountered by other researchers as well [see, for instance, A. Ralko *et al.* *Phys. Rev. B* 71, 224109 (2005).] In that paper, the authors showed that the phase transition point between the VBS and QSL depends on the cluster used and is about 0.7 ("type A cluster") or 0.85 ("type B cluster"). They also found it difficult to obtain the transition via traditional methods—such as looking at the order parameter or the Binder ratio—so they use the energy difference of different topological sectors to label the phase transition point. Since the topological sector is well-defined in the one-dimer-per-site limit, and the $\sqrt{12} \times \sqrt{12}$ VBS and QSL belong to different sectors in certain sizes, they finally obtained the phase transition point via this unconventional scheme. However, the phase transition between the $\sqrt{12} \times \sqrt{12}$ VBS and columnar phases is more difficult to calculate even through this way; what they instead claim is that "a clear columnar pattern appears near $V/t \sim -0.8$ ". Indeed, we also tried to repeat their result in our previous work [Z. Yan *et al.*,

npj Quantum Materials 6, 39 (2021)] in the one-dimer-per-site limit, but we also faced similar difficulties.

In the variable-dimer-density case, the problem becomes even harder. As the topological sector is no longer well-defined, we cannot compare the energy difference of the sectors. What we have tried is comparing the energy difference of the QMC results with different initial states. Over a large region, we found that the result will converge to the columnar/ $\sqrt{12} \times \sqrt{12}$ VBS phase if we set the initial state to be the columnar/ $\sqrt{12} \times \sqrt{12}$ VBS, but their energy difference is smaller than the error bar; thus, we cannot conclusively distinguish whether there is a columnar solid or $\sqrt{12} \times \sqrt{12}$ VBS.

Reviewer 2: 2. In the section on the dynamical dimer spectra, where the authors report the minima at the points M and Gamma in the odd- Z_2 liquid, it would be worth comparing with the similar spectrum from ref.[35] of the manuscript (A.Ralko et al, 2006), where it was studied for the $h=0$ case.

3. Further in this section, the authors mention earlier works on a single-vison dispersion in the even- Z_2 liquid. For a more balanced presentation, it would also be worth mentioning earlier works on a single-vison dispersion in the odd- Z_2 liquid in the $h=0$ case, specifically [D.Ivanov PRB 70, 094430 (2004)] – at the RK point $V=t$ and [A.Ralko et al, PRB 76, 140404(R) (2007)] – away from the RK point.

Reply: We thank the reviewer for their comments. In the revised version of the manuscript, we have now made suitable modifications as per these suggestions.

REVIEWERS' COMMENTS

Reviewer #1 (Remarks to the Author):

I appreciate the author's detailed response to my first report. They have satisfactorily addressed/ explained all points and questions I had.

Therefore I recommend publication of their manuscript in Nature Communications.

Reviewer #2 (Remarks to the Author):

The authors have satisfactorily addressed the questions in my first report, and I can now recommend the paper for publication.